# Characteristic Microbiomes Correlate with Polyphosphate Accumulation of Marine Sponges in South China Sea Areas

**DOI:** 10.3390/microorganisms8010063

**Published:** 2019-12-30

**Authors:** Huilong Ou, Mingyu Li, Shufei Wu, Linli Jia, Russell T. Hill, Jing Zhao

**Affiliations:** 1College of Ocean and Earth Science of Xiamen University, Xiamen 361005, China; hlou@xmu.edu.cn (H.O.); mingyuli@stu.xmu.edu.cn (M.L.); 22320151152141@stu.xmu.edu.cn (S.W.); 22320162201046@stu.xmu.edu.cn (L.J.); 2Institute of Marine and Environmental Technology, University of Maryland Center for Environmental Science, Baltimore, MD 21202, USA; 3Xiamen City Key Laboratory of Urban Sea Ecological Conservation and Restoration (USER), Xiamen University, Xiamen 361005, China

**Keywords:** sponge, polyphosphate, microbiomes, *ppk* gene

## Abstract

Some sponges have been shown to accumulate abundant phosphorus in the form of polyphosphate (polyP) granules even in waters where phosphorus is present at low concentrations. But the polyP accumulation occurring in sponges and their symbiotic bacteria have been little studied. The amounts of polyP exhibited significant differences in twelve sponges from marine environments with high or low dissolved inorganic phosphorus (DIP) concentrations which were quantified by spectral analysis, even though in the same sponge genus, e.g., *Mycale* sp. or *Callyspongia* sp. PolyP enrichment rates of sponges in oligotrophic environments were far higher than those in eutrophic environments. Massive polyP granules were observed under confocal microscopy in samples from very low DIP environments. The composition of sponge symbiotic microbes was analyzed by high-throughput sequencing and the corresponding polyphosphate kinase (*ppk*) genes were detected. Sequence analysis revealed that in the low DIP environment, those sponges with higher polyP content and enrichment rates had relatively higher abundances of cyanobacteria. Mantel tests and canonical correspondence analysis (CCA) examined that the polyP enrichment rate was most strongly correlated with the structure of microbial communities, including genera *Synechococcus*, *Rhodopirellula*, *Blastopirellula,* and *Rubripirellula*. About 50% of *ppk* genes obtained from the total DNA of sponge holobionts, had above 80% amino acid sequence similarities to those sequences from *Synechococcus*. In general, it suggested that sponges employed differentiated strategies towards the use of phosphorus in different nutrient environments and the symbiotic *Synechococcus* could play a key role in accumulating polyP.

## 1. Introduction

Marine sponges are ubiquitous in marine environments and can filter large amounts of water and mediate nutrient transformations by virtue of respiring organic matter and facilitating the consumption and release of nutrients [1,2]. By pumping seawater, marine sponges filter bacteria and organic particles as a food source [3]. Numerous microorganisms colonize sponge tissues [4], and can comprise up to 40–60% of the sponge biomass [3,5]. These microbes are involved in many physiological functions, such as development, defense, etc. [6]. Sponge-associated microbial metabolism substantially affects the biogeochemical cycling of key nutrients like carbon, nitrogen, and phosphorus [7]. Previous studies have confirmed the presence of nitrogen and sulfur nutrient cycles mediated by the sponge-associated microbial community [8,9]. Phosphorus in open-ocean surface waters is generally present in biologically restricting concentrations [10]. A comprehensive view of the oceanic phosphorus (P) cycle is also of broad significance because of its roles, as dissolved reactive phosphate in the ocean (primarily HPO_4_^2−^) in oceanic primary productivity [11]. In marine phosphorus cycles, many marine invertebrates can accumulate and store phosphorus in different forms, like yellow phosphorus [12] phospholipids [13]. Zhang et al. [14] discovered a major role for sponges with regard to the marine phosphorus cycle, in which phosphorus is sequestrated in the form of polyphosphate by the microbial symbionts of marine sponges. The ability of bacterial symbionts in sponges to concentrate phosphorus and store it in the form of polyphosphate raises the possibility that sponges may play a role as a sink of phosphorus and keep themselves thriving under the extremely low nutrient conditions in the marine reef ecosystems. Polyphosphate (polyP) production and storage by sponge endosymbionts, the P distribution in other sponge reef ecosystems, particularly those in extremely low nutrient environments, and the specific symbiotic microbial community response to polyphosphate sequestration have rarely been studied.

Polyphosphate (polyP), made up of linear polymers of tens to hundreds of phosphates, is an important form of marine inorganic phosphate. In areas rich in phosphorus, polyP is generally considered to be a result of the large uptake of phosphorus by organisms [15]. In oligotrophic environments, cells of phytoplankton can accumulate polyP in response to the resupply of phosphorus [16]. PolyP is a storage form of phosphorus in organisms and acts as an energy reserve for the phosphorylation of sugars, nucleosides and proteins, and activation of precursors of fatty acids, phospholipids, peptides, and nucleic acids. PolyP can therefore, be important for the long-term survival of organisms under adverse conditions [17]. Inorganic polyP has been confirmed as essential for the growth and survival of most microbes [17,18,19]. Moreover, there is plenty of evidence that polyP occurring in eukaryotic cells performs numerous regulatory functions [20]. Polyphosphate kinase (*ppk*) is the most important enzyme in prokaryotes for the synthesis of polyphosphates. The ppk enzyme polymerizes phosphate monomers onto existing polyphosphate long chains [17,21]. The finding of abundant phosphorus in the form of polyP in three sponges from the Florida Keys [14] raises the interesting possibilities that polyP abundance in sponges may be related to phosphorus concentrations in the surrounding waters and may depend on the presence of specific symbionts.

In marine reef-ecosystems, the associated microorganisms play critical roles in the decomposition of organic matter, transformation of nutrients, degradation of pollutants, maintenance of ecosystem sustainability, and regulation of marine productivity and host growth [6,22]. In marine sponges, as one important component of the reef ecosystem, the participation of phosphorus (P) cycle and the response of the associated microbial community to polyP formation are poorly understood. Here, the polyP accumulation in different sponge species from marine environments with different nutrient regimes was tested quantitatively by spectral analysis and direct observation with confocal microscopy. The corresponding sponge-associated microbiome and polyphosphate kinase (*ppk*) genes were investigated and analyzed with the aim of improving our understanding of the role of sponges in phosphorus sequestration and recycling in the reef environment. Our objective was to answer the following questions: (i) do all marine sponges have corresponding metabolic capabilities to accumulate inorganic phosphorus (mainly HPO_4_^2−^ and PO_4_^3−^) to form polyP? (ii) do different nutrient environments have an effect on the accumulation of P and polyP formation in marine sponges? (iii) are the associated microbiome and characteristic microbial groups correlated with polyP transformation?

## 2. Materials and Methods

### 2.1. Sample Collection and Site Description

A total of 12 sponge species (36 species individuals) were collected from the South China Sea (Appendix A; Table 1). Three species were sampled from Dongshan Bay, Zhangzhou City, Fujian Province, China in October 2016. This was a marine aquaculture area with mass nutrient flows in which sponges were abundant but limited in species. Nine species were sampled from the coast of Changhua town, Linqiangshi Island, Qizhou Island, and Meixia Port, Hainan Province, China, in September 2017. The diversity of marine sponges was relatively higher but the abundance was low. Dominant and subdominant sponge species were collected from each site; the number of individuals from the same species in a 20 m^2^ area was recorded as an estimate of the relative abundance of the sponges. Three replicates of each sponge species were collected. Tissue samples were cut into small pieces (approximately 1 cm^3^), and each sponge piece was rinsed three times with artificial seawater. Sponge specimens were identified according to the types of spicules and the morphology of the skeleton. The status of taxa was identified based on Systema Porifera and the World Porifera Database. For DNA extraction, the samples were stored in 70% (vol/vol) ethanol and preserved at −20 °C until use. For microscopic observation, the samples were fixed with 4% (wt/vol) paraformaldehyde at room temperature for 2 h and rinsed with 20 mM Tris buffer (pH 7.0) before being stored in 70% (vol/vol) ethanol and preserved at −20 °C until use. In addition, three 50 mL samples of seawater filtered through 0.45 μM filter membranes were collected from each sample location for dissolved inorganic phosphate (DIP) measurements at the same time as sponge collection and preserved at −20 °C until use. DIP in seawater was determined by using Auto Analyzer3 (AA3, Bran+Lubbe, D-22844, Norderstedt, Germany).

### 2.2. Extraction and Measurement of PolyP from Sponge Samples

The extraction and measurement of polyP generally followed the “core” protocol described by Martin and Van Mooy [23]. First, lyophilized sponge tissue (~20 mg) from three individuals of each species was finely ground and dissolved in 20 mM Tris buffer (pH 7.0) (Sangon Biotech Co., Ltd, Shanghai, China), then homogenized by vortexing. The suspension was sonicated for 1 min and immersed in boiling water for 10 min. Then, the samples were treated with 40 units/mL DNase and 400 units/mL RNase at 37 °C for 30 min to remove DNA and RNA, followed by 20 mg/mL proteinase K. Later, samples were centrifuged at 15,000 *g* for 2 min. Next, supernatants were collected and 200 μL of supernatant from each extraction was transferred to 96 well array plates, and DAPI was added to a final concentration of 10 μM, followed by incubation at room temperature for 5 min. Fluorescence measurements were carried out on a Tecan Infinite M200 Pro fluorometer. The excitation wavelength was set to 415 nm; emission spectra were recorded from 450–620 nm in 5-nm increments. The remaining pellets were subjected to additional successive extractions following the same protocol until the fluorescence of 535 nm (fluorescence peak of polyP in biological samples) moving to 485 nm. A polyP standard (S4379, Sigma-Aldrich, 200216, Shanghai, China) was used for quantification. Aliquots were dissolved in 20 mM Tris buffer (pH 7.0). On binding of DAPI to polyp, there is a shift in the emission wavelength to a peak of 550 nm on excitation at 405 nm. This peak fluorescence at 550 nm can be used to quantify the concentration of polyP by comparison to the polyP standard curve (Appendix A). The mass of polyP was calculated from the polyP concentration, the volume of the lysate and molecular weight of the polyP standard. To determine the polyP content, the polyP mass was divided by the sponge mass. DIP in the water from the five sampling sites was determined using a Nutrient Salt Automatic Analysis System. To evaluate the accumulation ability of sponges for polyP, the enrichment rate was defined as the ratio of polyP in sponge tissue to DIP in the surrounding water.

### 2.3. Visualization of PolyP Granules by Confocal Microscopy

The sponge tissue (approximately 0.3 cm^3^) was washed for 10 min in each of a series of solutions: 70% ethanol, 80% ethanol, 90% ethanol, 100% ethanol, 50% xylene/ethanol (vol/vol), and 100% xylene and embedded in paraffin. Next, 10 μm thick paraffin sections were cut, placed on microscope slides, dewaxed in xylene and ethanol respectively, and washed three times in MilliQ water. The sections were stained with 20 μM DAPI solution for 2 min, washed with MilliQ water three times and visualized under a Zeiss LSM780NLO (Carl Zeiss AG, 73447, Oberkochen, Germany) inverted confocal microscope equipped with a 20 × Axio Observer Z1 automatic inverted fluorescence/lens. Sample sections were visualized by excitation with a 405-nm laser light source with emission signals separated by a NFT515 filter into two channels. Emission wavelength from 420–515 nm resulting from DAPI bound to nucleotides was collected in Channel 1. The emission wavelength above 530 nm, including the peak of 550 nm resulting from DAPI bound to polyP, was collected in Channel 2. The polyP standard curve was shown in Appendix A.

### 2.4. DNA Extraction and 16S rDNA PCR Amplification and Sequencing

The metagenomic DNA was extracted directly from the hosts following the methods of Zhou et al. [24]. For each sample, DNA was extracted in triplicate to minimize bias, then the three extracts from each sample were pooled. DNA purity and concentration were analyzed using a spectrophotometer (Nanodrop 2000, Thermo Fisher Scientific, 02451, Waltham, MA, USA) and agarose gel electrophoresis and the DNA was stored at −20 °C until use. The primers 515F (5′-GTGCCAGCMGCCGCGGTAA-3′) and 806R (5′-GGACTACHVGGGTWTCTAAT-3′) [25], which target the hypervariable V4 region of the 16S rRNA gene, were used for amplification under the following conditions: 94 °C for 5 min; 30 cycles of 94 °C for 30 s, 55 °C for 30 s, and 72 °C for 1 min; and a final extension at 72 °C for 10 min. Amplicons were visualized by gel purification. Bands were excised from gels and purified by ethanol precipitation prior to Illumina HiSeq 2500 PE250 Platform sequencing.

### 2.5. Amplicon Sequencing Data Processing and Analysis

To get high-quality clean reads, raw reads were filtered according to the following rules: (1) reads containing more than 10% unspecified (N) nucleotides were discarded; (2) reads containing less than 80% bases with low-quality values (Q-value > 20) were discarded. The remaining paired-end reads were merged as raw tags using the program FLASH [26] (v 1.2.11). The criteria used were a minimum overlap of 10 bp and a requirement for mismatch error rates below 2%. The QIIME [27] (V1.9.1) pipeline was used as described in reference [28] to obtain high-quality clean reads. Sequences were searched against the reference database (http://drive5.com/uchime/uchime_download.html, access on 9/8/2018) to perform reference-based chimera checking which was performed by searching against a reference database (http://drive5.com/uchime/uchime_download.html, access on 9/8/2018) using UCHIME (http://www.drive5.com/usearch/manual/uchime_algo.html, access on 9/8/2018). Chimeric reads were removed and the remaining reads were used in subsequent analyses. These sequences were clustered into operational taxonomic units (OTUs) of  ≥97% similarity by using the UPARSE [29] pipeline. A single high-abundance sequence was selected as the representative sequence within each cluster. Venn analysis between groups was performed in R. The organisms from which the representative sequences were derived was deduced by a naive Bayesian model using the RDP classifier [30] (Version 2.2) based on the SILVA [31] database (https://www.arb-silva.de/, access on 12/12/2018). The abundance statistics were used to create taxonomic groups and phylogenetic trees constructed with a Perl script and visualized using SVG. Chao1, Simpson and all other alpha diversity indices were calculated in QIIME. OTU rarefaction curves and rank abundance curves were plotted in QIIME. The alpha index comparison between groups was calculated by a Welch’s t-test and a Wilcoxon rank test in R (https://www.r-project.org). Alpha index comparison among groups was computed by a Tukey’s HSD test and a Kruskal-Wallis H test in R. To examine the relationship between the microbial community structure and sponge abundance, DIP, polyP and the enrichment rate, canonical correspondence analyses (CCA) were conducted using the CANOCO software [32]. The Mantel test was performed to assess the correlations between microbial communities and environmental factors by using the R package “vegan” [33].

The sequence data were deposited in the National Center for Biotechnology Information (NCBI) GenBank Sequence Read Archive under accession numbers SAMN10713642 to SAMN10713671.

### 2.6. PCR Amplification and Cloning of Polyphosphate Kinase Genes from Sponges

Genomic DNA was extracted as mentioned above. The developed *ppk* primers amplified two *ppk* gene fragments with lengths of approximately 1000 bp and 600 bp, respectively. Two pairs of primers targeting *ppk* gene were designed by web-based Primer-Blast of the NCBI (National Center for Biotechnology Information) and by BIO-edit software and Primer Premier 5.0 based on main microbial phyla with sponges in this study (Appendix A). PCR products purified by a universal DNA purification kit were ligated into the pMD19-T vector and transformed into DH5α chemically competent *Escherichia coli* cells using the pMDTM19-T Vector Cloning Kit, then selected by monoclonal colony PCR. The *ppk* gene libraries were constructed and 50–100 sequences were randomly selected and were sequenced by Invitrogen Corporation.

Sequences of the *ppk* gene obtained in this study were deposited in the GenBank database under accession numbers MK119939-MK119959, MK119965-MK119968, and MK119974-MK119978.

## 3. Results

### 3.1. Determination of PolyP Concentrations in Sponges from Different Marine Environments

Sampling sites in the South China Sea were divided into high DIP and low DIP areas based on the value in water samples and location (Table 1). Dongshan Bay was a eutrophic environment with high DIP (~4.796 μM) near which there was a large aquaculture area. Nevertheless, the amounts of polyP accumulated by sponge in Dongshan Bay were significantly different, with 0.024 ± 0.001 mg/g detected in *Callyspongia* sp. (DC), and 0.008 ± 0.004 mg/g detected in *Mycale* sp. (DM) which was the lowest in all samples detected, while as much as 3.722 ± 0.152 mg/g was detected in *Tedania* sp. (DT). In other marine environments (Changhua Town, Linqiangshi Island, Meixia Port, and Qizhou Island), the concentration of DIP was lower by ~2 orders of magnitude compared to Dongsan Bay. In Changhua Town, DIP in seawater was only ~0.040 μM (Table 1). However, a large amount of polyP was accumulated by *Haliclona* sp. (CH01, 3.729 mg/g) and *Cladocroce* sp. (CH02, 2.911 mg/g). CH01 contained the highest polyP concentration in all the sponges tested. PolyP accumulation within the same sponge genus from different locations showed distinct variations, for example, *Mycale* sp. LQ02 from the Linqiangshi Island area, in which the seawater DIP level (~0.210 μM) was far below that in Dongshan Bay (~4.796 μM), accumulated relatively more polyP (LQ02, 0.137 mg/g) than DM (0.008 mg/g). In contrast, the same species *Callyspongia* sp. in high or low DIP marine areas stored the similar levels of polyP, DC samples with 0.024 ± 0.001 mg/g and MX03 with 0.046 ± 0.019 mg/g.

Statistical data (*p* < 0.05, Pearson Test) supported that the polyP accumulation ability of each sponge could be positively related to the abundance and growth characteristics of sponge species (Table 1). At the same site, more dominant sponge individuals with flourishing growth generally were found to accumulate more polyP. In Qizhou Island, the abundance of *Sigmaxinella* sp. QZ01 was higher than that of *Ircinia* sp. QZ02, and QZ01 accumulated polyP up to 1.355 mg/g, compared to 0.977 mg/g in QZ02. The relative enrichment rates of sponge species in the low DIP environment were generally higher than those in high DIP areas. The enrichment rates of CH01, CH02, LQ01, MX01, MX02, QZ01, and QZ02 were over 10^5^, while enrichment rates in eutrophic environments (especially DM and DC) were much lower. Sponge *Mycale* sp. LQ02 and *Callyspongia* sp. MX03 showed a relatively higher enrichment rate (31215 ± 9399, 13121 ± 5455) in the oligotrophic environment than *Mycale* sp. DM (53 ± 27) and *Callyspongia* sp. DC (157 ± 10) in the eutrophic environment, sponge genera *Mycale* and *Callyspongia* presented the least absorption and conversion of polyP compared with other sponges. Conversely, sponges (CH01, CH02) sampled near Changhua Town with the lowest seawater DIP showed the highest polyP content per gram sponge tissue and the highest relative enrichment rates.

### 3.2. PolyP Distribution by Confocal Microscopy

To independently confirm the presence of polyP, sections of sponge tissue were visualized by fluorescence microscopic imaging with DAPI staining. When DAPI-polyP was excited at 405 nm, it emitted a typical green-yellow fluorescence signal in cells that represented the accumulation of high levels of polyP, while lower levels of polyP were not detectable by fluorescence microscopic imaging (Figure 1). Massive polyP granules with green fluorescence were observed in samples CH01, CH02, MX01, and QZ01 which had relatively higher polyP levels and enrichment rates (Figure 1A–D). In contrast, sponges with a lower polyP accumulation ability showed few or no granules in the blue cell background (Figure 1E,F).

### 3.3. Sponge Microbiomes

A total of 1,724,653 sequence reads were obtained from 36 samples by 16S rRNA high-throughput sequencing. Eukaryotes-derived sequences were excluded in the subsequent analyses. 21,058 operational taxonomic units (OTUs) were assigned to 313 known genera. Rarefaction curves with OTUs at 97% similarity indicated that most samples showed great microbial diversity. There were obvious differences in the sequences coverage for observed species, ranging from 31% to 86% of the diversities predicted by estimation of expected species, but most samples were recovered at 60–79% (Appendix A). Alpha-diversity indices of different samples (Appendix A) showed that the microbial richness and diversity were significantly different in various sponge species from the same collection sites (CH01-vs.-CH02, LQ01-vs.-LQ02, and MX01-vs.-MX02-vs.-MX03), and in the same sponge genus from different collection sites (LQ02-vs.-MX01 and MX02-vs.-QZ02).

The microbial diversity was inextricably linked to momentous changes with the sponge host and the local environment ecosystem. The sample *Cladocroce* sp. CH02 in an oligotrophic area exhibited the most diverse bacterial community, with 43 phyla, and *Mycale* sp. DM from a eutrophic environment inversely harbored the fewest microbial phyla (20). In Dongshan Bay with high DIP, the percentage of Proteobacteria (53%) and Cyanobacteria (38%) associated with sponge *Tedania* sp. DT was more than 90%, while Proteobacteria (79%) and Actinobacteria (12%) were the main microbial groups with *Mycale* sp. DM from the same location. Although there were other populations with relative abundance >1% found in sponges, Cyanobacteria, Proteobacteria, and Planctomycetes were the common and dominant microbial groups in nearly all sponges inhabiting low DIP marine environments, except *Ircinia dendroides* MX02 and *Callyspongia* sp. MX03. In Meixia Port, Cyanobacteria, Thaumarchaeota, Proteobacteria, and Planctomycetes were the dominant microorganisms associated with *Lissodendoryx* sp. MX01 (>80%) and *Callyspongia* sp. MX03 (>90%), while Cyanobacteria, Chloroflexi, Nitrospinae, Acidobacteria, and PAUC34f were abundant in *Ircinia dendroides* MX02 (>70%). In the same sponge genera *Ircinia* sp. QZ02 from Qizhou Island, Cyanobacteria, Chloroflexi, Proteobacteria, Planctomycetes, and Nitrospinae were the dominant microorganisms (88%) (Figure 2). On the other hand, the principal coordinates analysis (PCoA) from 12 sponge samples showed the sponge-associated microbiome from the same sites could cluster closely, but other symbionts with the same genus sponges (DT vs. LQ02, DC vs. MX03) presented the distinct community structure (Appendix A).

Though the bacterial compositions at the class level were similar among the sponges, they showed marked differences in the richness of bacterial taxa. Cyanobacteria were the most abundant bacterial group in almost all sponge samples (16–81%), except that it was the second-most abundant group in sponge sample QZ02 (7%) and had a low abundance in DM (0.7%). In Changhua Town sponges, the abundance of cyanobacteria in *Haliclona* sp. CH01 (79%) was higher than that in *Cladocroce* sp. CH02 (50%). Especially, *Synechococcus* in CH01 (9%) was about 10 times higher than that in CH02 (0.9%) (Appendix A). In Linqiangshi Island, the abundance of cyanobacteria in *Lissodendoryx* sp. LQ01 (63%) was higher than that in *Mycale* sp. LQ02 (42%). In Meixia Port, the abundance of cyanobacteria in MX01 (58%) was higher than that in MX02 (16%). In Qizhou Island, sponge *Sigmaxinella* sp. QZ01 was rich in cyanobacteria (81%), while *Ircinia* sp. QZ02 was rich in Caldilineae (Chloroflexi, 47%).

Mantel tests and canonical correspondence analysis (CCA) were performed to examine the relationships between microbial community structures and various aspects, including DIP, sponge abundance, polyP content, and enrichment rates (Figure 3 and Figure 4). The Mantel test indicated that all of the four examined factors were correlated with microbial distribution (Figure 3; *p* < 0.05). Among all of the factors examined, the polyP enrichment rate (the longest arrow) was most strongly correlated with the structure of microbial communities (Figure 4). In particular, the abundance of *Synechococcus*, *Blastopirellula*, *Rhodopirellula,* and *Rubripirellula*, which were among the top 20 genera with sponges (Appendix A), had the strongest correlation with the polyP enrichment rate. Relative to the enrichment rate and DIP, sponge abundance (r = 0.308, *p* = 0.002) and polyP content (r = 0.391, *p* = 0.001) were secondarily correlated with microbial community structures (Figure 4).

### 3.4. ppk Gene Identification and Host-Specific Microbial Groups in Relationship to polyP Sequestration

Polyphosphate kinase (*ppk*) plays an important role in polyP synthesis in prokaryotes. The polyphosphate kinase gene encodes the enzyme that catalyzes polyP production. By constructing gene libraries, a total of 30 sequences from *ppk* gene libraries were identified from eight sponge holobionts (Table 2), indicating the existence of the *ppk* genes in the sponge samples. The results of BLASTX search showed that *ppk* was present in six main bacterial taxa: the Proteobacteria (α-, β-, γ-), Cyanobacteria, Actinobacteria, and Acidobacteria. Among them, one sequence was closely related to *ppk* from α-proteobacteria. Nine sequences were related to *ppk* from γ-proteobacteria, specifically Xanthomonadales, Oceanospirillales, Enterobacterales, Thiotrichales, and Acidiferrobacterales. Notably, 15 sequences presenting about 50% *ppk* gene libraries from DM, DT, CH01, CH02, LQ01, LQ02, and QZ01 were highly homologous to those of *Synechococcus* (Cyanobacteria) with above 80% amino acid similarity. In the sponge *Mycale* sp. DM, one of nine *ppk* gene sequences was assigned to *Synechococcus* species, while in the other six sponge samples except for *Ircinia* sp. QZ02, nearly 100% of the *ppk* sequences had the closest similarities to those from *Synechococcus* species.

Compositions of the microbial structure were explored through 16S rDNA amplicon sequencing. In Changhua Town, Linqiangshi Island, Qizhou Island, and Meixia Port, sponges with high polyP enrichment rates showed higher abundances of *Synechococcus* than other sponges from the same area (Figure 4, Appendix A), while sponges with low polyP enrichment rates from Dongshan Bay (DM and DT) had low abundances of *Synechococcus* genus.

## 4. Discussion

Consideration of the distribution of inorganic nutrients and sponge reefs in the world’s oceans suggests that reefs can be present in water with quite a wide range of nutrient levels but are generally found in oligotrophic waters [34]. In this study, sponges were flourishing in both oligotrophic and eutrophic areas with low or high inorganic phosphorus levels. Although there were no clear patterns in polyP sequestration and utilization, different sponges could employ different strategies towards phosphorus. In the high DIP environment, *Tedania* sp. DT seemed to use polyP but not store enough polyP granules to be observed by confocal microscopy, while *Mycale* sp. DM could adopt other ways to utilize phosphorus but not polyP. In contrast, sponges existing in those areas with very low DIP could maintain requisite nutrients by enhancing their abilities of polyP accumulation and forming polyP granules, such as *Haliclona* sp. CH01, *Cladocroce* sp. CH02, *Sigmaxinella* sp. QZ01, and *Lissodendoryx* sp. MX01. The high enrichment rates and the storage of phosphorus as polyP may have significance for sponges living in oligotrophic environments by protecting sponges from phosphorus limitation and enhancing survival in a marine desert. A similar phenomenon occurred in the Sargasso Sea when phytoplankton accumulated polyP in response to low phosphorus levels [16]. The fact that more abundant sponges contained more polyP within the same regions, may imply that these sponges have a selective advantage in phosphorus accumulation that enables them to become more dominant [35].

In marine ecosystems, phosphorus concentration is generally controlled through external abiotic processes, making it difficult to balance supply and demand through biological activities [36,37]. In view of the wide distribution of sponges and the diversity of their symbiotic microorganisms, polyP synthesis in sponge symbiotic microorganisms and the process of biomineralization in sponges may have a major impact on the phosphorus cycle in benthic ecosystems [14]. In this study, canonical correspondence analysis (CCA) revealed the strong correlation between the polyP enrichment rate and the microbiomes of sponges. Microbial symbionts can make crucial contributions to host metabolism. Although host phylogeny and host identify influenced symbiotic microbial taxa and their related metabolic pathways to a great extent [37], under selection for environmental pressure, the effect of host phylogeny was reduced, such as the same genera *Mycale* sp. (DM vs. LQ02) or *Callyspongia* sp. (DC vs. MX03) in high/low DIP environment (Appendix A). A group of microorganisms referred to as polyphosphate (polyP) -accumulating organisms (PAOs) are usually involved in biological phosphorus transfer [38]. If PAOs form the dominant microbial community, this may contribute to the lower diversity of the microbial community structure [39]. Combined with polyP accumulation, PAOs, mainly cyanobacteria but also Planctomycets and Proteobacteria and to a less extent, Acidobacteria, could be related to phosphorus sequestration.

Among those, cyanobacteria-associated sponges with high polyP enrichment rates were more abundant than those with low polyP enrichment rates and low polyP contents. Traditionally, the role of symbiotic cyanobacteria in sponges has been considered to be mainly related to the nutrient supply to their hosts through their photosynthetic and nitrogen-fixing activities; carbon fixation and nitrogen fixation have been widely reported to be related to symbiotic cyanobacteria [40,41]. Cyanobacteria can provide up to 50% of the energy source and 80% of the carbon source for tropical sponges through photosynthesis [42]. In contrast, less attention has been given to the relationship between symbiotic cyanobacteria and the phosphorus cycle in the host. The finding that cyanobacteria can accumulate polyphosphate from water as a storage form of phosphorus [43,44,45] and the recent discovery of polyP granules in three coral reef sponges and their symbiotic cyanobacteria in the Caribbean [14] raise the possibility that symbiotic cyanobacteria may also have a crucial role in phosphorus cycling.

PolyP from symbiotic microorganisms accounted for 25–40% of the total phosphorus of sponges, which indicated that microorganisms could mediate the chelation of phosphorus in sponges [1]. In this study, by combining microbial community structure and CCA analysis, a possible link between *Synechococcus* genus (clustered in Cyanobacteria) and the polyP enrichment rate was established. The composition of polyphosphate kinase (*ppk*) genes from gene libraries of sponge holobionts, further implied that the microbial sources of polyP in sponges at the genetic level [14]. From the diversity and percentage, nearly half of *ppk* genes shared a high homology with *Synechococcus*; it was suggested that *Synechococcus* could synthesize polyphosphate as one of the energy reserves to supply the host sponge in oligotrophic environments. Previous studies on polyphosphate accumulation mostly focus on free-living cyanobacteria [46,47]. In 2009, studies on marine cyanobacteria by Bily et al. [48] showed that *Synechococcus elongatus* could store excess phosphate and synthesize polyphosphate particles in the cytoplasm or vacuoles to remove phosphorus in the water. Multiple groups of Synechococcales were observed with transmission electron microscopy to relate to polyP particles found in bacterial cells [49,50], confirming that Synechococcales stored polyP in response to phosphorus limitation in the environment [51].

Our previous study about *Tedania* sp. from embryo to adult stages showed that *Synechococcus* occurred in sponge species as horizontal acquisitions but not inherited symbionts, even though they were not abundant in surrounding seawater [52]. These microorganisms were specifically taken up from seawater and formed functional symbioses. Under conditions of P abundance and deficiency, these symbionts are likely to participate in the phosphorus metabolism of the host via synthesis and decomposition of polyP. In the ocean, symbiotic algae of coral [53] and symbiotic bacteria of bivalves [54] had been reported to accumulate polyP under no-stress conditions and to degrade polyP as an energy supply to maintain the metabolic activity of their hosts under environmental stress. In this study, polyP particles were observed in sponge tissues, related to polyP conversion and highly abundant cyanobacteria were detected, which supported the conjecture that symbiotic microorganisms could be positively related to the polyP accumulation of sponge hosts. In the future, cyanobacteria, particularly *Synechococcus* and more sponge species should be further studied by omics in regard to P synthesis, P storage, and P release mechanisms in sponges. This would be an important step in understanding the role of sponges and their symbionts in phosphorus cycling in sponge-dominated marine reefs.

## Figures and Tables

**Figure 1 microorganisms-08-00063-f001:**
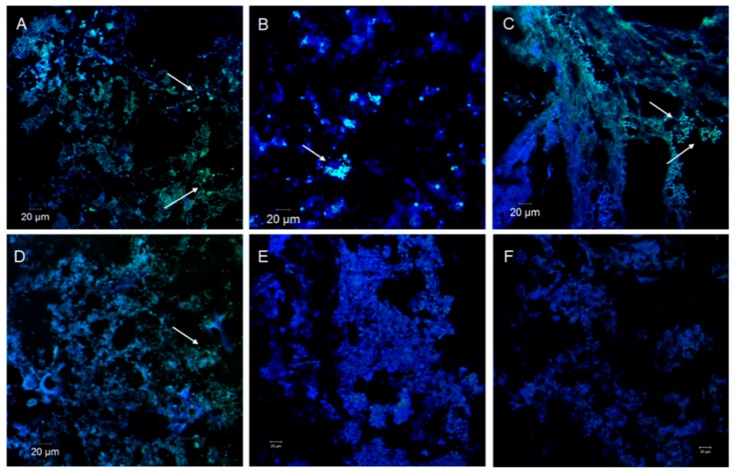
Visualization of polyP in sponge tissues by DAPI-staining under fluorescence microscopic imaging. PolyP granules indicated by arrows. (**A**). CH01, (**B**). CH02, (**C**). MX01, (**D**). QZ01, (**E**). DT, (**F**). DM. Samples in (**A–D**): from the oligotrophic environment; Samples in (**E**,**F**): from the eutrophic environment. Scale bar: 20 μm.

**Figure 2 microorganisms-08-00063-f002:**
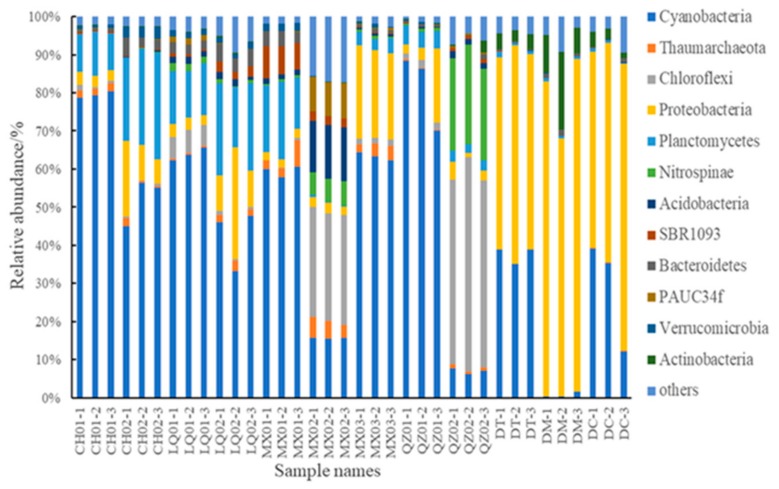
The microbial community composition and abundance patterns in the level of phyla of different taxonomic groups from 36 sponge samples. CH01: *Haliclona* sp., CH02: *Cladocroce* sp., LQ01: *Lissodendoryx* sp., LQ02: *Mycale* sp., MX01: *Lissodendoryx* sp., MX02: *Ircinia* dendroides, MX03: *Callyspongia* sp., QZ01: *Sigmaxinella* sp., QZ02: *Ircinia* sp., DT: *Tedania* sp., DM: *Mycale* sp., DC: *Callyspongia* sp.

**Figure 3 microorganisms-08-00063-f003:**
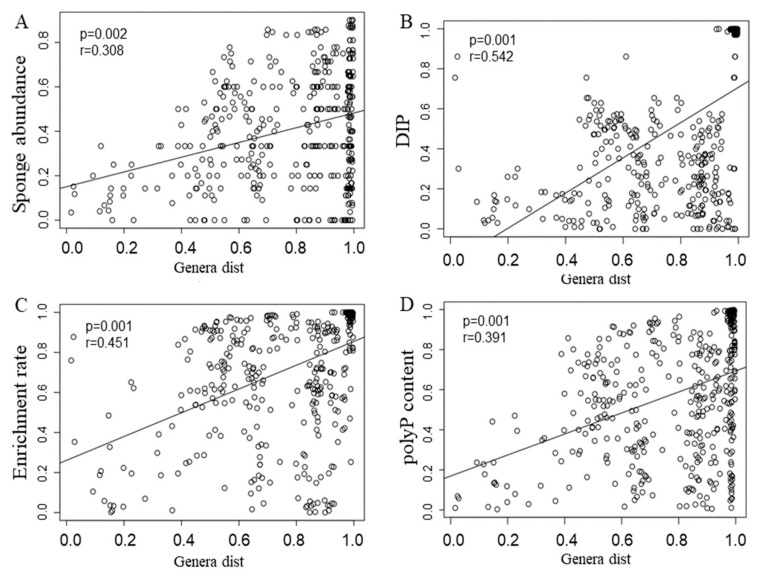
Mantel Test on the relationship between bacterial diversity at the level of genus and environmental factors. The X-axis presented a distance metric of taxonomic composition. (**A**) sponge abundance; (**B**) Dissolved inorganic phosphorus; (**C**) enrichment rate of polyphosphate; (**D**) polyphosphate content of sponge.

**Figure 4 microorganisms-08-00063-f004:**
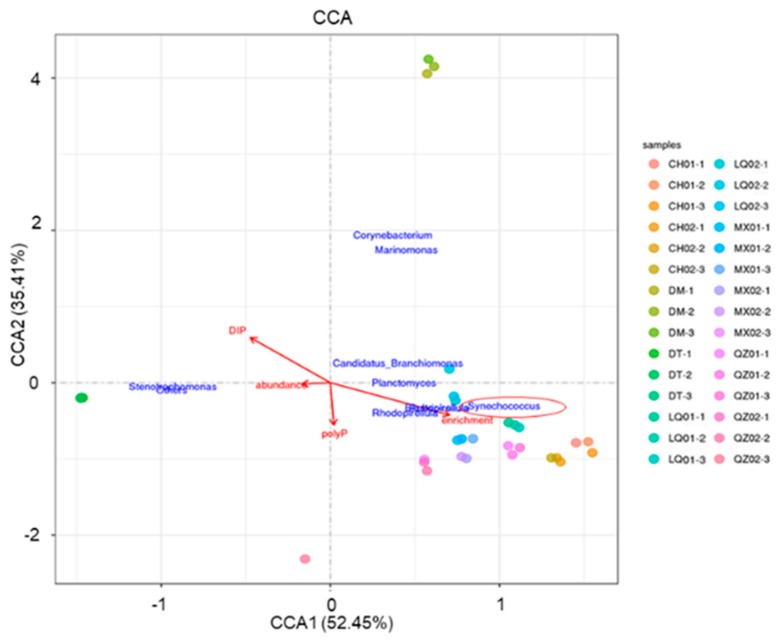
Canonical correspondence analysis (CCA) between the microbial community structure and four environment factors.

**Table 1 microorganisms-08-00063-t001:** Phosphorus sequestration in the form of polyphosphate by sponges in oligotrophic and eutrophic marine environments (*n* = 3 for each sample).

Sample Sources	DIP ^1^ in Surrounding Sea Water (mM)	ID	Sponge Taxonomy	Abundance (Individuals/20 m^2^) ^3^	polyP/Sponge Dry Weight (mg/g)	Enrichment Rate ^2^
Dongshan bay (E) 23.80° N, 117.59° E	~4.796	DM	*Mycale* sp.	5	0.008 ± 0.004	53 ± 27
		DT	*Tedania* sp.	16	3.722 ± 0.152	24813 ± 1807
		DC	*Callyspongia* sp.	10	0.024 ± 0.001	157 ± 10
Changhua Town (O) 19.25° N, 109.03° E	~0.040	CH01	*Haliclona* sp.	12	3.729 ± 1.711	3107893 ± 247702
		CH02	*Cladocroce* sp.	7	2.911 ± 0.133	2425581 ± 111206
Linqiangshi lsland (O) 19.53° N, 109.26° E	~0.210	LQ01	*Lissodendoryx* sp.	4	0.484 ± 0.079	110015 ± 17986
		LQ02	*Mycale* sp.	2	0.137 ± 0.041	31215 ± 9399
Qizhou Island (O) 19.55° N, 111.11° E	~0.081	QZ01	*Sigmaxinella* sp.	3	1.355 ± 0.161	541971 ± 64371
		QZ02	*Ircinia* sp.	1	0.977 ± 0.619	390672 ± 247702
Meixia Port (O) 20.00° N, 109.35° E	~0.114	MX01	*Lissodendoryx* sp.	4	1.159 ± 0.443	331027 ± 126645
		MX02	*Ircinia dendroides*	2	0.492 ± 0.194	140559 ± 55423
		MX03	*Callyspongia* sp.	3	0.046 ± 0.019	13121 ± 5455

^1^ DIP means dissolved inorganic phosphorus; ^2^ Enrichment rate was defined as the ratio of polyP in per gram sponge tissue to DIP in surrounding sea environment; (E) means eutrophic sites, (O) means oligotrophic sites; ^3^ The correlation analysis of polyP and sponge species abundance with Pearson test: *p* < 0.041.

**Table 2 microorganisms-08-00063-t002:** The BLASTX results of *ppk*1 gene.

ID	Closest *ppk* Relative and Its Accession Number	Bacteria Group	AA Identities	Taxonomy
DM.1	polyphosphate kinase 1 WP_045783506.1	*Klebsiella michiganensis*	97%	Gammaproteobacteria; Enterobacterales
DM.2	polyphosphate kinase KRO79340.1	OM182 bacterium BACL3 MAG-120920-bin41	96%	Gammaproteobacteria; OMG group
DM.3	polyphosphate kinase 1 WP_049475655.1	*Stenotrophomonas maltophilia*	99%	Gammaproteobacteria; Xanthomonadales
DM.4	polyphosphate kinase 1 WP_071965591.1	*Streptomyces cinnamoneus*	73%	Actinobacteria; Streptomycetales
DM.5	polyphosphate kinase KPK47330.1	Thiotrichales bacterium SG8_50	72%	Gammaproteobacteria; Thiotrichales
DM.6	polyphosphate kinase 1 OYV98205.1	Acidobacteria bacterium 21-70-11	72%	Acidobacteria
DM.7	polyphosphate kinase 1 WP_019874751.1	*Sporichthya polymorpha*	77%	Actinobacteria; Frankiales
DM.8	polyphosphate kinase KRO79340.1	OM182 bacterium BACL3 MAG-120920-bin41	96%	Gammaproteobacteria; OMG group
DM.9	polyphosphate kinase 1 PWL23716.1	*Synechococcus* sp. XM-24	86%	Cyanobacteria; Synechococcales
DT.1	polyphosphate kinase 1 WP_045783506.1	*Klebsiella michiganensis*	98%	Gammaproteobacteria; Enterobacterales
DT.2	polyphosphate kinase 1 WP_010309567.1	*Synechococcus* sp. CB0101	84%	Cyanobacteria; Synechococcales
DT.3	polyphosphate kinase 1 WP_049475655.1	*Stenotrophomonas maltophilia*	98%	Gammaproteobacteria; Xanthomonadales
DT.4	polyphosphate kinase 1 PWL23716.1	*Synechococcus* sp. XM-24	89%	Cyanobacteria; Synechococcales
DT.5	polyphosphate kinase 1 OYV98205.1	Acidobacteria bacterium 21-70-11	71%	Acidobacteria.
DT.6	polyphosphate kinase 1 WP_094558932.1	*Synechococcus* sp. 8F6	86%	Cyanobacteria; Synechococcales
CH01.1	polyphosphate kinase 1 WP_010309567.1	*Synechococcus* sp. CB0101	86%	Cyanobacteria; Synechococcales
CH01.2	polyphosphate kinase 1 WP_007099397.1	*Synechococcus* sp. RS9916	97%	Cyanobacteria; Synechococcales
CH01.3	polyphosphate kinase 1 WP_010317776.1	*Synechococcus* sp. CB0205	85%	Cyanobacteria; Synechococcales
CH01.4	polyphosphate kinase 1 PWL23716.1	*Synechococcus* sp. XM-24	84%	Cyanobacteria; Synechococcales
CH02.1	polyphosphate kinase 1 WP_007099397.1	*Synechococcus* sp. RS9916	97%	Cyanobacteria; Synechococcales
CH02.2	polyphosphate kinase 1 WP_041025907.1	*Alcanivorax pacificus*	84%	Gammaproteobacteria; Oceanospirillales
LQ01.1	polyphosphate kinase 1 WP_010309567.1	*Synechococcus* sp. CB0101	87%	Cyanobacteria; Synechococcales
LQ01.2	polyphosphate kinase 1 PWL23716.1	*Synechococcus* sp. XM-24	88%	Cyanobacteria; Synechococcales
LQ01.3	polyphosphate kinase 1 WP_010309567.1	*Synechococcus* sp. CB0101	89%	Cyanobacteria; Synechococcales
LQ02.1	polyphosphate kinase 1 WP_010309567.1	*Synechococcus* sp. CB0101	89%	Cyanobacteria; Synechococcales
QZ01.1	polyphosphate kinase 1 WP_007099397.1	*Synechococcus* sp. RS9916	97%	Cyanobacteria; Synechococcales
QZ01.2	polyphosphate kinase 1 PWL23716.1	*Synechococcus* sp. XM-24	87%	Cyanobacteria; Synechococcales
QZ02.1	polyphosphate kinase PPR24019.1	Alphaproteobacteria bacterium MarineAlpha10_Bin2	74%	Alphaproteobacteria
QZ02.2	polyphosphate kinase 1 WP_090634088.1	*Nitrosomonas marina*	74%	Betaproteobacteria; Nitrosomonadales
QZ02.3	polyphosphate kinase 1 WP_096458022.1	*Sulfurifustis variabilis*	65%	Gammaproteobacteria; Acidiferrobacterales

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
