# Peer review of "Characteristic Microbiomes Correlate with Polyphosphate Accumulation of Marine Sponges in South China Sea Areas"

_microorganisms, 2019, doi:10.3390/microorganisms8010063_

Round 1
Reviewer 1 Report
The authors present a survey of polyP accumulation in some regional sponges, try to statistically correlate polyP levels to bacterial OTU distribution, and then search for polyP biosynthesizing ppk genes in sponge metagenomes. The study design has the reviewer pose the following questions:
the ppk gene primer design is not presented. How do the authors validate that the ppk gene amplification primers used in this study are not biased towards prokaryotes, and specifically, towards cyanobacteria? Do the gene libraries constructed detect any eukaryotic ppk genes? A more robust analysis would have been a retro-transcriptomic analysis. Even though ppk gene is located in the cyanobacteria, it may be meaningless if the gene is not expressed, or, if the eukaryotic ppk gene, which is currently not detected due to flawed primer design, is expressed at a very high level. Though it is tantalizing to suggest that the cyanobacteria make polyP, the current study design not allow for this assertion to made with high confidence.2. Syncechococcus are exceptionally abundant in other sponge biomes as well. Do all sponges that harbor Synechococcus make polyP? Most definitely not. Hence, the correlation fails when viewed in that light. The authors should describe their findings in that light.
3. The number of sponge members analyzed here seems to be rather small. Within each polyP enriched sponges, numerous samples should have been analyzed to observe if the cyanobacterial OTUs hold up at similar abundance levels.
4. Maybe I missed this, but, do the authors describe what is the amplicon length generated for the ppk genes and if that amplicon length is sufficient for phylogenetic differentiation? BLAST similarity does not mean much if the amplicon length is too small and not validated to correspond to phylogenetic differences.
5. The reviewer would suggest remaking Figures 2-4 for clarity, at the moment, the features are too small to see clearly.
Reviewer 2 Report
This is a very interesting article that covers an emerging and very important topic in coral reef ecology. My comments are relatively minor:
Methods:
How were the sponges identified?
There were three replicates of water samples collected, but these were collected at the same time. Is there any evidence that DIP levels change over time? This could impact the enrichment rates. Could sponges act as a more integrated assessment of ambient DIP levels?
Results:
There should be some statistical support for some of the claims here. For instance, on likes 213 to 219 the "positively related" statement, and statements about relative enrichment rates. These might be significant, but all important results should be tested to make sure that these trends really are there.
Lines 252 to 267 and 271 to 285: Is all of this summary important for the overall story? Could this be tested statistically in a more robust way? It seems like there is an effect of host identity, site, etc on dissimilarity across these microbiome samples. Could this be tested to assess this effect-like Easson and Thacker, 2014. This might make for a more concise story here.
Figure 2: I recommend picking one of these figures to highlight and moving the other to supplemental. Also, please add species names instead of abbreviations for readers.
Lines 293 to 295: These are still significant relationships. I don't think you need to say that they are not clearly correlated.
Round 2
Reviewer 1 Report
The revisions are well done the reviewer recommends publication.
Author Response
The manuscript has been carefully checked and corrected.